# Serum IgG4 Concentration Is a Potential Predictive Biomarker in Glucocorticoid Treatment for Idiopathic Retroperitoneal Fibrosis

**DOI:** 10.3390/jcm11123538

**Published:** 2022-06-20

**Authors:** Shoichiro Mukai, Naotaka Sakamoto, Hiroaki Kakinoki, Tadamasa Shibuya, Ryosuke Moriya, Kiyoaki Nishihara, Mitsuru Noguchi, Toshitaka Shin, Naohiro Fujimoto, Tsukasa Igawa, Tatsu Ishii, Nobuhiro Haga, Hideki Enokida, Masatoshi Eto, Tomomi Kamba, Hideki Sakai, Seiichi Saito, Naoki Terada, Toshiyuki Kamoto

**Affiliations:** 1Department of Urology, Faculty of Medicine, University of Miyazaki, 5200 Kihara, Kiyotake-cho, Miyazaki 889-1692, Japan; naoki_terada@med.miyazaki-u.ac.jp (N.T.); urokamo@gmail.com (T.K.); 2Department of Urology, National Hospital Organization Kyushu Medical Center, Fukuoka 810-8563, Japan; sakamoto.naotaka.kf@mail.hosp.go.jp; 3Department of Urology, Faculty of Medicine, Saga University, Saga 849-8501, Japan; kakinoki@cc.saga-u.ac.jp (H.K.); nogman@cc.saga-u.ac.jp (M.N.); 4Department of Urology, Faculty of Medicine, Oita University, Yufu 879-5593, Japan; tadamasa@oita-u.ac.jp (T.S.); shintosh@oita-u.ac.jp (T.S.); 5Department of Urology, School of Medicine, University of Occupational and Environmental Health, Kitakyushu 807-8555, Japan; moriryo@med.uoeh-u.ac.jp (R.M.); n-fuji@med.uoeh-u.ac.jp (N.F.); 6Department of Urology, Kurume University School of Medicine, Kurume 830-0011, Japan; nishihara_kiyoaki@med.kurume-u.ac.jp (K.N.); tigawa@med.kurume-u.ac.jp (T.I.); 7Department of Urology, Fukuoka University Chikushi Hospital, Chikushino 818-8502, Japan; ishii-t@adm.fukuoka-u.ac.jp; 8Department of Urology, Faculty of Medicine, Fukuoka University, Fukuoka 814-0180, Japan; nhaga@fukuoka-u.ac.jp; 9Department of Urology, Faculty of Medicine, Kagoshima University, Kagoshima 890-8520, Japan; henokida@m2.kufm.kagoshima-u.ac.jp; 10Department of Urology, Graduate School of Medical Sciences, Kyushu University, Fukuoka 812-8582, Japan; etom@uro.med.kyushu-u.ac.jp; 11Department of Urology, Faculty of Life Sciences, Kumamoto University, Kumamoto 860-8556, Japan; kamba@kumamoto-u.ac.jp; 12Department of Urology, Nagasaki University Graduate School of Biomedical Sciences, Nagasaki 852-8501, Japan; hsakai@nagasaki-u.ac.jp; 13Department of Urology, Graduate School of Medicine, University of the Ryukyus, Nishihara 903-0215, Japan; ssaito@med.u-ryukyu.ac.jp

**Keywords:** idiopathic retroperitoneal fibrosis, glucocorticoid, immunoglobulin G4, predictive biomarker

## Abstract

Objectives: To evaluate the management and outcome of idiopathic retroperitoneal fibrosis (iRPF) in Japan, and to identify its clinical biomarker. Methods: We retrospectively analyzed 129 patients with iRPF treated between January 2008 and May 2018 at 12 university and related hospitals. Patients treated with glucocorticoid were analyzed to identify a predictive biomarker. These patients were classified into three groups according to overall effectiveness (no change: NC, complete response: CR, and partial response groups: PR), and each parameter was compared statistically. Results: Male–female ratio was 5:1, and median age at diagnosis was 69 (33–86) years. Smoking history was reported in 59.6% of the patients. As treatment, 95 patients received glucocorticoid therapy with an overall response rate of 84%. As a result, serum concentration of IgG4 was significantly decreased in NC group compared with the other two groups (56.6 mg/dL vs. 255 mg/dL, 206 mg/dL, *p* = 0.0059 and 0.0078). ROC analysis was performed between the nonresponder (NC) and responder groups (CR + PR) to identify the cut-off value of serum IgG4 as a predictive marker. As a result, AUC of 0.793 was confirmed. Conclusions: Pre-treatment serum IgG4 concentration may have potential as a predictive biomarker of steroid treatment.

## 1. Introduction

Retroperitoneal fibrosis (RPF) is characterized by the presence of fibrotic tissue in retroperitoneum, accompanying chronic inflammation, fibrotic tissue surrounding the abdominal aorta and iliac arteries, and ureteral obstruction [1,2,3,4,5]. The presence of fibrotic tissue in retroperitoneum is common in all cases, whereas accompanying chronic inflammation, fibrotic tissue surrounding the abdominal aorta and iliac arteries, and ureteral obstruction are observed to varying degrees in individual cases. Because of its rarity and benign characteristics, accepted evidence seems to be lacking. Therefore, epidemiology, classification, and etiology as well as treatment procedures and information on outcome remain elusive, and this has acted as an obstacle to the establishment of guidelines [1,2,3,4,5]. Published reports estimate the incidence of RFP to be 0.1–1.3/100,000 persons per year, a male–female ratio of 2:1–3:1, and a mean age at diagnosis ranging between 55 and 60 years [3,4,5,6,7,8]. 

RPF has been classified into secondary and idiopathic type [1,2,3,4,5]. Secondary RPF includes drug-induced and infectious-disease-related etiologies. There is evidence regarding the relationship between ergot alkaloids or dopamine agonists and RPF [3,9]. In addition, surgery or radiation therapy for retroperitoneal lesions also induce fibrotic change, which may cause secondary RPF [1,2,3,4,5]. On the other hand, 70–80% of RPF is classified as idiopathic type, which includes immunoglobulin G4-related disease (IgG4RD) and IgG4-unrelated chronic aortitis, inflammatory aortic aneurysm, and perianeurysmal fibrosis [1,2,3,4,5,6]. Although glucocorticoid therapy shows a good response rate in idiopathic RPF (iRPF), predictive biomarkers for the therapy have yet to be established. 

Because the rarity of this disease makes it difficult to collect an adequate number of patients at a local institution, we planned collaborative research with twelve universities and eight related hospitals in the Kyushu–Okinawa area to collect data on RPF patients. In the present report, we retrospectively evaluated clinical data including etiology, management and the outcome of the RPF patients, and discussed serum IgG4 concentration as a predictive biomarker in glucocorticoid therapy. 

## 2. Materials and Methods

### 2.1. Patients

The present multi-institutional study was conducted with the approval of all institutional review boards (approval No. O-0336, 2018). We retrospectively analyzed patients with RPF treated between January 2008 and May 2018 at 12 university hospitals and related hospitals. Major criteria were presence of fibrotic tissue in retroperitoneal area without regard to the presence or absence of accompanying inflammation or hydronephrosis. No apparent primary retroperitoneal tumor or retroperitoneal metastasis were observed. Malignant neoplasm occurring in locations other than the retroperitoneal area was included in iRPF. Surgical intervention, radiation to the retroperitoneal area and exact drug-induced RPF (ergot alkaloids and dopamine agonists) were defined as secondary RPF (Figure 1). Clinical data including age, gender, comorbidity, past history, results of laboratory examination, appearance of imaging examination, urological examination, details of treatment and the outcome were retrospectively extracted from clinical records.

### 2.2. Baseline Measurements

Baseline laboratory examination, including complete blood cell count, serum level of creatinine, C-reactive protein (CRP), soluble interleukin-2 receptor (sIL-2R), IgG and IgG4, were performed during the pre-treatment period. Plaque size was analyzed by computed tomography (CT) and/or magnetic resonance imaging (MRI). Due to the various shapes and locations of the plaque, we decided to use maximum diameter for the determination of size. Measurable cases were analyzed for treatment outcome. 

### 2.3. Evaluation of Treatment Outcome

Mass reduction rate was calculated using the changing value of maximum diameter {(pretreatment-posttreatment)/pre-treatment}. Response of medical treatment was judged as follows: reduction rate of >15% for “reduction” (especially, >90% as “complete reduction”); 0–15% as “no reduction”. 

Improvement of urinary tract was evaluated by degree of obstruction (hydronephrosis) in upper urinary tract and condition of ureteral stent. Response was classified into the following four groups: “stent free from the beginning”; “became stent free”; “improved but still stenting”; “no improvement”. The first two groups were judged as “improved”, and others were no “improvement”.

Overall effect of treatment was analyzed both by mass reduction rate and improvement of urinary tract, and then it was classified into three groups of “complete response: CR”, “partial response: PR”, and “no change: NC”. Patients classified into CR include both complete reduction and stent-free, PR contains reduction and/or improved, and NC consists of both no reduction and no improvement. 

### 2.4. Statistical Analysis

This is a fact-finding study without control cohort. Therefore, the result is aggregate data with limited statistical analysis. Intergroup differences were analyzed using the Mann–Whitney *U*-test (StatFlex version 7, Artech, Osaka, Japan) to identify a predictive biomarker of glucocorticoid therapy. A P value of less than 0.05 was considered statistically significant. Receiver operating characteristic (ROC) analysis was also performed using StatFlex version 7, and cut-off value was identified by the Youden index. 

## 3. Results

During the study period, a total of 145 RPF patients were treated at each hospital, and a total of 129 patients were classified as idiopathic RPF (iRPF). Initial patient characteristics are shown in Table 1. As a result, there were 108 male patients with a male–female ratio of 5.1:1.0. Median age at diagnosis was 69 (33–86) years. Smoking history was reported in 59.6% of the patients. Review of comorbidity and past history revealed autoimmune disease in 14.3% (including pancreatitis, thyroiditis, Sjoegren’s disease, idiopathic thrombocytopenic purpura, polymyositis, uveitis, rheumatoid arthritis, psoriasis vulgaris and cholangitis), allergy in 8.6%, nonretroperitoneal malignant neoplasm in 13.4% and arteriosclerotic disease in 32.8%. Abdominal surgery not within the retroperitoneal area was performed in 18.1% of the patients (transurethral resection of bladder tumor was included in iRPF). One patient received radiation therapy not within the retroperitoneal area. No patients were treated with ergot alkaloids or dopamine agonists. 

The breakdown of treatment administered to the 127 patients (two patients had no information for treatment) was glucocorticoid in 95 cases, Saireito in 6 cases, mizoribine in 1 case, and two agents (glucocorticoid and rituximab) in 1 case. Surgical intervention was performed in 7 cases, and 17 patients received no treatment (only urinary tract management and observation). 

We focused on glucocorticoid therapy, which is a major treatment of iRPF. A total of 94 patients with evaluable outcome were analyzed. As a result, reduction of plaque size was observed in 79 (84%) cases at the median observation period of 82 days (interquartile range: 34–120 days). Of interest is that complete reduction was observed in 13 cases treated by glucocorticoid. Treatment effect for urinary tract was analyzed in 89 cases. 

Among 58 patients with ureteral stent inserted prior to glucocorticoid treatment, 31 (53.4%) patients became stent free as a result of treatment. The overall response rate in this study was 84% (79/94). 

Reduction of plaque size was observed in six cases in the nontreatment group; however, details were unclear because of a very short observation period. Surgical intervention was performed in eight cases; however, ureteral injury occurred in two cases, and five cases continued to be managed by stenting after surgery. 

We then analyzed the parameters of patients that received glucocorticoid therapy to identify a predictive biomarker. As shown in Table 2, patients were classified into three groups according to overall effectiveness, and each parameter was compared statistically. Median reduction rates were NC: 2.5% (range 0–15%), CR: 99.0% (range 92–100%), PR: 47.42% (range 15.2–90). As a result, serum concentration of IgG4 was significantly decreased in the NC group compared with the other two groups (Figure 2, *p* = 0.0059 and 0.0078, respectively). 

ROC analysis was performed between nonresponder (NC, *n* = 11) and responder groups (CR and PR, *n* = 60) to identify the cut-off value of serum IgG4 as a predictive marker. ROC curve, cut-off value and area under the curve (AUC) are shown in Figure 3. As a result, the cut-off value was 67.6 mg/dL, and AUC was 0.793. The values of sensitivity and specificity were 0.85 and 0.636.

Elevation of antinuclear antibody was observed in 17.9% (7/39) of the patients (NC: 1, CR: 1, PR: 5); however no statistical difference was observed. Other autoantibodies were not analyzed in this study. 

Contents of glucocorticoid therapy is shown in Table 3. Initial treatment period of responder group, and maintenance treatment period of nonresponder group, seemed to be higher; however, no significant differences were noted for each group.

## 4. Discussion

In this study, we collaborated with 12 university and 8 related hospitals to collect clinical data on 144 RPF patients. The study was a retrospective investigation; therefore, statistical analysis was problematic due to missing values (data) as well as differences among hospitals in diagnostic and treatment strategies, and evaluation of outcomes. However, recent real RPF treatment data in the Kyushu–Okinawa area was available, allowing this retrospective study to include the largest number of Japanese patients to date. 

Diagnosis of RPF was based on imaging studies performed by attending physicians in each hospital with or without pathological diagnosis. The primary diagnostic finding for inclusion was presence of fibrotic tissue in retroperitoneal lesion. Presence of ureteral obstruction or inflammation were not required for inclusion. 

In our study, mean age at diagnosis was slightly higher than that published in previous reports [2,3,4,5,6,7]. Although this was not a population-based study, male predominance was apparent with a male–female ratio of 5.1:1.0. The result was higher than that found in the literature [2,3,4,5,6,7]. As described below regarding risk factor of RPF, high incidence of smoking history (56.5%, 61/108) and arteriosclerotic disease (35.2%, 38/108) in male RPF patients may cause this (male predominance). A significant association between RPF and smoking has been reported. In this case-control study, the risk was higher (odds ratio of 3.21) in current smokers and former smokers (odds ratio of 2.93) compared with individuals with no history of smoking [10]. Indeed, a high incidence (59.6%) of smoking history was observed in our study, and the result is consistent with previous reports. RPF has been reported to include chronic periaortitis, perianeurysmal fibrosis and inflammatory abdominal aortic aneurysm, which suggests a meaningful correlation between arteriosclerotic disease and RPF [2,3,4,5,6]. Compared with previous studies, the incidence of arteriosclerotic disease as a comorbidity was high (32.8%), and the incidence of autoimmune disease was similar (14.3%) in the current study [2,3,4,5,6]. In addition, results of 17 patients who were administered aspirin and beta-adrenergic blocker may correlate with arteriosclerotic disease. 

Outcome of iRPF in reduction of size and/or urinary tract improvement was evaluated for a total of 126 patients in this study. As a result, glucocorticoid therapy achieved a favorable response rate (84%), and ureteral stent was removed in 53.4% of the patients. The response rate was slightly higher than previous reports [2,3,4,5,6]. The results indicate the apparent efficacy of glucocorticoid, and that glucocorticoid therapy is a standard first-line treatment for iRPF. As the next step, standardization of treatment protocol is necessary. In the literature, the recommended treatment strategy is an initial dose of 0.6–1 mg/kg/day of prednisolone for 2–4 weeks, with the dosage gradually tapered to 2.5–5 mg/day over a period of greater than 6 months [2]. Tanaka et al. suggested the convenience of treatment initiated with 0.6 mg/kg/day of prednisolone for 4 weeks, and then a 10% reduction in dosage every 2 weeks for IgG4-related disease (IgG4RD) [5]. Indeed, median initial dose was 30 mg for 26.5 days in the current study, and a total of 46% of the patients started with 30 mg/day of prednisolone. The results of our study were similar to the protocol for IgG4RD, and the efficacy seemed to be acceptable. No significant differences were noted between responder group and nonresponder group in treatment dose and period in both initial and maintenance phase. However, the maintenance treatment period in nonresponder group seemed to be longer than in responder group, which suggests unnecessarily prolonged treatment. A prospective study analyzing both efficacy and safety would be helpful in establishing a standard protocol for iRPF.

Recently, the majority of idiopathic RPF has been classified as IgG4RD, and greater than half of idiopathic RPF patients were reported to have been diagnosed with IgG4RD by histological examination [4]. However, the exact pathophysiology is controversial due to the fact that fundamental IgG4RD is a systemic disease, whereas RPF occurs in a limited area of the retroperitoneum. Frequency of concurrent RPF in IgG4RD has been reported as 3–19% [11], and approximately 60% of RPF is reported to be associated with IgG4RD [12]. However, no systematic analysis of a large cohort to evaluate the association with IgG4RD has been conducted. In the current study, only 7 cases were diagnosed as IgG4RD by pathological analysis; however, a total of 69 patients were analyzed for serum IgG4 concentration, and elevation (cut-off value of <125 mg/dL) was observed in 43 patients (62.3%). 

According to the 2020 Japanese revised comprehensive diagnostic criteria for IgG4RD, seven cases were diagnosed as “definite” for IgG4RD, and 36 patients are “probable” for IgG4RD (retroperitoneal plaque and elevation of serum IgG4 concentration, cut-off value of >135 mg/dL) [13]. Therefore, the possibility of IgG4RD being misdiagnosed as iRPF remains. Although the treatment strategy is similar between iRPF and IgG4RD, pathological finding is needed for a definitive diagnosis. 

Interestingly, patients with high serum IgG4 concentration showed a favorable response to glucocorticoid therapy. In spite of the limitation of our study, statistical significance was observed. Although the exact number of potentially overlapped IgG4RD could not be evaluated, serum IgG4 concentration may be a candidate as a predictive biomarker of glucocorticoid therapy. In this study, acceptable AUC (0.793) was observed, and cut-off value was calculated as 67.6 mg/dL. However, the values of sensitivity and specificity (0.85 and 0.636) might not be sufficient as cut-off levels. Further prospective examination is recommended to clarify our result. 

A tendency for high serum IgG and low sIL-2R concentration was also observed in the favorable response group; however, no statistical significance was observed. Serum sIL2-R was examined to rule out malignant lymphoma. As a result, increased serum level was observed in 74.5%; however, the degree of increase was not significant compared with that of malignant lymphoma [14]. RPF is also associated with inflammation. In addition, infiltration of both B cells (and/or IgG4-positive plasma cell) and T cells has also been confirmed pathologically. Therefore, the phenomenon may not be a disease-specific result; however, our investigation revealed that this is the first time that elevated serum sIL2-R level has been described in RPF.

Saireito is a traditional herbal medicine (Kampo medicine) with an anti-inflammatory effect and is used for treatment of RPF in Japan [15]. Favorable efficacy for RPF without severe adverse event has been seen in several case reports; however, exact cohort data such as response rate are lacking. Mizoribine is an imidazole nucleoside [16]. Similar to rituximab, the agent was also used for RPF expecting an immunosuppressive effect. 

As mentioned above, limitations of our study are its being a retrospective investigation, which created obstacles to accurate statistical analysis due to missing values (data), and differences among hospitals in diagnostic and treatment strategies as well as evaluation of outcomes. 

## 5. Conclusions

We conducted a fact-finding study on 129 iRPF patients treated in the Kyushu–Okinawa area of Japan. The majority of iRPF patients received glucocorticoid therapy and achieved a favorable response. As a predictive biomarker of steroid treatment, pre-treatment serum IgG4 concentration may be useful. The establishment of acceptable guidelines is strongly recommended. 

## Figures and Tables

**Figure 1 jcm-11-03538-f001:**
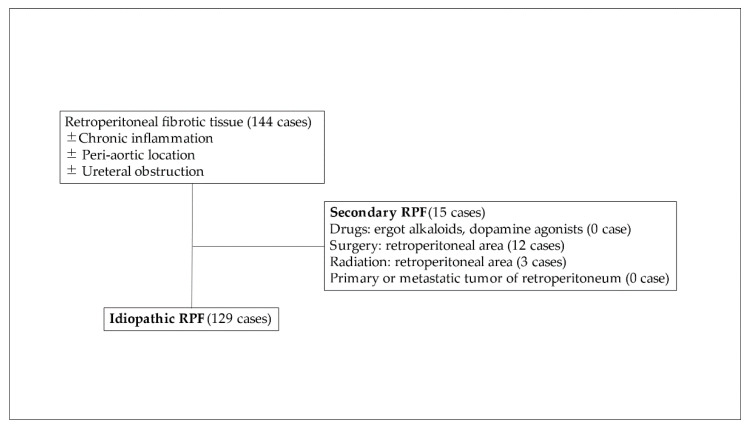
Classification of retroperitoneal fibrosis, and selection of study cohort. A total of 144 cases were enrolled, and 15 cases were classified as secondary RPF. Patients with idiopathic RPF were analyzed in this study.

**Figure 2 jcm-11-03538-f002:**
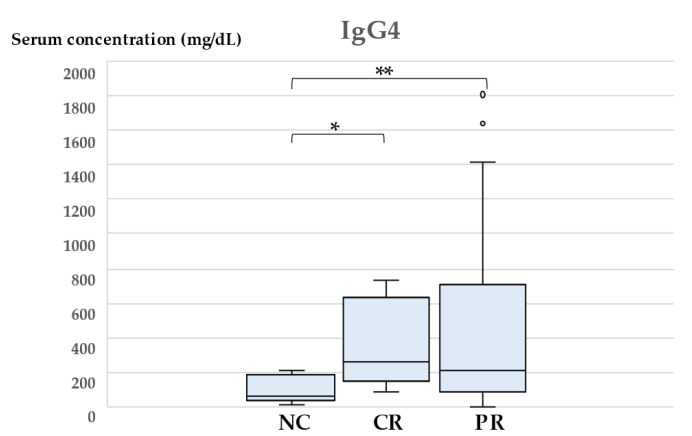
Serum concentration of IgG4 in each group classified by response to glucocorticoid therapy. Intergroup differences were analyzed using the Mann–Whitney *U*-test. A *p* value < 0.05 was considered statistically significant. IgG4, Immunoglobulin G4; sIL2-R, soluble interleukin-2 receptor; NC, no change; CR, complete response; PR, partial response. * *p* = 0.0059; ** *p* = 0.0078.

**Figure 3 jcm-11-03538-f003:**
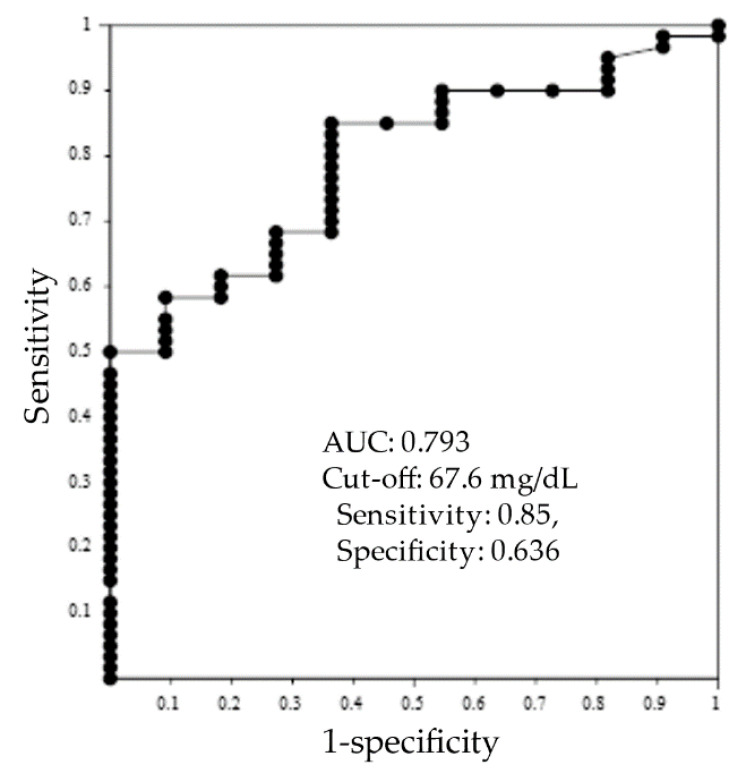
Receiver operating characteristic (ROC) curve analysis of IgG4levels. The area under the ROC curve and cut-off value were calculated using responder group (*n* = 60) and nonresponder group (n = 11) data. AUC, area under the curve.

**Table 1 jcm-11-03538-t001:** This is a ta.

	*n*/Total	%
Age at diagnosis, years (range)	69 (33–86)	
Gender		
Male	108	83.7
Female	21	16.3
Smoking history	62/104	59.6
Asbestos exposure	3/69	4.3
Comorbidity and past history		
Autoimmune disease	18/126	14.3
Allergy	11/128	8.6
Malignant neoplasm	17/127	13.4
Arteriosclerotic disease	42/128	32.8
Abdominal surgery	23/127	18.1
Radiation therapy	1/126	0.8
Treatment		
Glucocorticoid	95/127	74.8
Saireito	6/127	4.7
Other agents	2/127	1.6
Surgical intervention	7/127	5.5
No treatment	17/127	13.4
Overall effect (glucocorticoid therapy)		
complete response	13/94	13.8
partial response	66/94	70.2
no change	15/94	16
Stent free rate	31/58	53.4

**Table 2 jcm-11-03538-t002:** Laboratory data in each group classified by response to glucocorticoid therapy.

	NC			CR			PR		
	Median (Range)	Mean ± SD	*n*	Median (Range)	Mean ± SD	*n*	Median (Range)	Mean ± SD	*n*
Age, years	65 (51–81)	65.3 ± 8.9	15	65 (50–79)	66.1 ± 7.6	13	65 (50–79)	68.7 ± 8.9	66
WBC (×10^9^/L)	6815 (4980–13,490)	7637 ± 2378	14	6900 (4900–11,100)	7139 ± 2159	13	6750 (2490–16,700)	7143 ± 2289	66
Hb (g/dL)	12.7 (8.9–14.1)	12.07 ± 1.78	15	11.9 (8–15.5)	12.2 ± 2.43	13	12.45 (8.6–18.7)	12.48 ± 1.9	66
Cre (mg/dL)	1.55 (0.68–14.03)	3.26 ± 4.32	15	1.08 (0.56–10.98)	1.83 ± 2.79	13	1.3 (0.59–14.2)	2.1 ± 2.59	66
CRP (mg/dL)	0.91 (0.05–8.89)	3.07 ± 3.72	12	0.51 (0.03–8.04)	1.61 ± 0.69	13	1.37 (0.03–18.94)	2.46 ± 3.57	65
sIL2–R (U/mL)	1038.5 (243–1840)	1061.8 ± 526.1	10	888 (500–1335)	899.4 ± 271	9	819 (372–2361)	1027.7 ± 560.7	42
IgG (mg/dL)	1613 (921.6–3268)	1679.1 ± 620.7	12	1727 (982–4590)	2085.1 ± 1198.4	7	1714 (859–3857)	2013.1 ± 753	43
IgG4 (mg/dL)	56.6 (3.3–209)	93.8 ± 75.9	10	255 (81–729)	355.4 ± 251.2	8	206 (2–1810)	386.2 ± 437.5	51

**Table 3 jcm-11-03538-t003:** Contents of glucocorticoid therapy.

	No Responder	Responder	*p*-Value
Initial treatment	*n* = 15	*n* = 79	
Median dose (IQR), mg/day	30 (26.25–30)	30 (25–40)	0.239
Median period (IQR), days	21 (14–49.3)	28 (14–35)	0.938
Maintenance treatment	*n* = 11	*n* = 63	
Median dose (IQR), mg/day	5 (2.5–5)	5 (5–5)	0.471
Median period (IQR), days	696 (135–979)	210 (111.8–720)	0.232

## Data Availability

The datasets used and/or analyzed during the current study are not publicly available due to identifiable patient information but are available from the corresponding author on reasonable request.

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
