# Peer review of "Serum IgG4 Concentration Is a Potential Predictive Biomarker in Glucocorticoid Treatment for Idiopathic Retroperitoneal Fibrosis"

_jcm, 2022, doi:10.3390/jcm11123538_

Round 1
Reviewer 1 Report
Shoichiro Mukai et al. presented an interesting report on serum IgG4 concentration as a potential predictive biomarker in glucocorticoid treatment of idiopathic RPF.
Idiopathic retroperitoneal fibrosis is a rare pathology, and I appreciate the effort involved in producing such a report, even if it is only retrospective.
The majority of iRPF patients received glucocorticoid therapy, and most achieved a favorable response. As a predictive biomarker for steroid treatment, serum IgG4 concentration could be useful according to the results of this study.
I think the authors probably got as much as they could out of the data available to them, considering they used data from more than 10 centers across the country.
However, I think a few things need to be better articulated:
-First, please describe in more detail the measurement of the extent of RPF and the associated definition of criteria for improvement after therapy.
-In the introduction or discussion, the possible treatments need to be briefly described. The authors mention agents such as saireito and misoribine, which may not be familiar to many readers.
-The NC group had higher IL2R levels. Although the differences are not statistically significant, we invite comment. Do those with a stronger inflammatory component respond better to therapy?
Author Response
Thank you very much for your important comments regarding our paper. We revised the manuscript according to the comments. We hope that the reviewer will be satisfied with our response.
-First, please describe in more detail the measurement of the extent of RPF and the associated definition of criteria for improvement after therapy.
Response: Thank you very much. According to the suggestion, we added comments in Materials and Methods (page 3, line 99-104) and Results (page 7, line 11-12).
-In the introduction or discussion, the possible treatments need to be briefly described. The authors mention agents such as saireito and misoribine, which may not be familiar to many readers.
Response: According to the comments, we added comments in Discussion (page 10, line 136-140) and references Nos. 14-15.
-The NC group had higher IL2R levels. Although the differences are not statistically significant, we invite comment. Do those with a stronger inflammatory component respond better to therapy?
Response: We appreciate for the reviewer’s important suggestion. We added comments in Discussion (page 10, line 127-135) and reference No.13.
Reviewer 2 Report
Mukai et al. report that the serum IgG4 level is a potential predictive biomarker for treatment response in idiopathic retroperitoneal fibrosis. This study is a retrospective design. However, the authors show the clinical importance and usefulness of IgG4 levels in predicting treatment responses. I have a minor comment on this study.
1. Only seven patients have pathologic diagnoses for IgG4-related disease in this study. However, more than half of the patients (69 patients) are elevated serum IgG4 levels. Considering the difficulties of the diagnostic approach to a retroperitoneal area, is there any possibility of missed diagnoses for IgG4-related diseases? The authors need a more detailed discussion about the relationship between iRPF and IgG4RD.
2. Why is there male predominance? The authors need to address these results in the discussion section.
3. What autoimmune diseases are associated with iRPF in comorbidity and past medical history? The authors need a detailed explanation in the discussion section and add the information about autoantibodies such as serum antinuclear antibody levels in table 2 if available data.
Author Response
Thank you very much for your important comments regarding our paper. We revised the manuscript according to the comments. We hope that the reviewer will be satisfied with our response.
- Only seven patients have pathologic diagnoses for IgG4-related disease in this study. However, more than half of the patients (69 patients) are elevated serum IgG4 levels. Considering the difficulties of the diagnostic approach to a retroperitoneal area, is there any possibility of missed diagnoses for IgG4-related diseases? The authors need a more detailed discussion about the relationship between iRPF and IgG4RD.
Response: We appreciate for the reviewer’s important suggestion. We added comments in Discussion (page 9, line 113-page 10, line118) and reference No.16.
- Why is there male predominance? The authors need to address these results in the discussion section.
Response: Thank you very much. According to the suggestion, we revised and added comments in Discussion (page 9, line 70-72).
- What autoimmune diseases are associated with iRPF in comorbidity and past medical history? The authors need a detailed explanation in the discussion section and add the information about autoantibodies such as serum antinuclear antibody levels in table 2 if available data.
Response: According to the suggestion, we added comments in Results (page 3, line 130-132 and page 7, line 20-22).
